# Synthesis and Functionalisation of Superparamagnetic Nano-Rods towards the Treatment of Glioblastoma Brain Tumours

**DOI:** 10.3390/nano11092157

**Published:** 2021-08-24

**Authors:** Kinana Habra, Stéphanie E. B. McArdle, Robert H. Morris, Gareth W. V. Cave

**Affiliations:** 1School of Science and Technology, Nottingham Trent University, Nottingham NG11 8NS, UK; kinana.habra2017@my.ntu.ac.uk (K.H.); rob.morris@ntu.ac.uk (R.H.M.); 2John van Geest Cancer Research Centre, School of Science and Technology, Nottingham Trent University, Nottingham NG11 8NS, UK; stephanie.mcardle@ntu.ac.uk

**Keywords:** carnosine, glioblastoma, hyperthermia, hydrothermal, iron oxide nano-rods, cancer, U87 MG cells, MRI, branched polyethyleneimine

## Abstract

The complete removal of glioblastoma brain tumours is impossible to achieve by surgery alone due to the complex finger-like tentacle structure of the tumour cells and their migration away from the bulk of the tumour at the time of surgery; furthermore, despite aggressive chemotherapy and radiotherapy treatments following surgery, tumour cells continue to grow, leading to the death of patients within 15 months after diagnosis. The naturally occurring carnosine dipeptide has previously demonstrated activity against in vitro cultured glioblastoma cells; however, at natural physiological concentrations, its activity is too low to have a significant effect. Towards realising the full oncological potential of carnosine, the dipeptide was embedded within an externally triggered carrier, comprising a novel nano rod-shaped superparamagnetic iron oxide nanoparticle (ca. 86 × 19 × 11 nm) capped with a branched polyethyleneimine, which released the therapeutic agent in the presence of an external magnetic field. The new nano-carrier was characterized using electron microscopy, dynamic light scattering, elemental analysis, and magnetic resonance imaging techniques. In addition to cytotoxicity studies, the carnosine carrier’s effectiveness as a treatment for glioblastoma was screened in vitro using the U87 human glioblastoma astrocytoma cell line. The labile carnosine (100 mM) suppresses both the U87 cells’ proliferation and mobility over 48 h, resulting in significant reduction in migration and potential metastasis. Carnosine was found to be fully released from the carrier using only mild hyperthermia conditions (40 °C), facilitating an achievable clinical application of the slow, sustained-release treatment of glioblastoma brain tumours that demonstrates potential to inhibit post-surgery metastasis with the added benefit of non-invasive monitoring via MRI.

## 1. Introduction

Glioblastoma multiforme (GBM) is the most malignant form of glioma and accounts for about 15% of all brain tumours and 60% of primary brain tumours. The prognosis for GBM patients, even after surgical ablation of the tumour and subsequent chemo-radiation therapy, is poor, with a mean life expectancy of less than 14.6 months [1]. The main contributing factor to the low survival rates is attributed to the persistence of residual malignant cells that have been found to be resistant to chemotherapy and radiotherapy, resulting in rapid recurrence [2]. It is, therefore, imperative to look towards new theragnostic approaches to treat this terminal disease [1,3]. In order to mitigate the risk of residual malignant cells post-surgery, real time medical imaging (e.g., MRI) during surgery is emerging as a new tool for surgeons to help reduce recurrence; however, to optimise this process, a targeted contrast agent is required [4,5,6,7].

Chelated gadolinium is used in over 30% of all MRI scans as a contrast agent; however, its widespread clinical use is limited [8]. Despite being fully chelated (typically with EDTA analogues) [9], gadolinium ions are reported to leach into the blood stream, resulting in nephrogenic systemic fibrosis and signs of neurological disorders [10]. Super paramagnetic iron oxide, SPIO, nanoparticles are, therefore, emerging as clinically approved alternatives [11]. This can be attributed to their lower toxicity and colloidal stability at physiological conditions, which should be always optimised to prevent cases of severe reactions and blood clots [12,13], controllable surface charge and their low nonspecific protein adsorption [14,15]. The combined medical imaging and potential therapeutic properties of SPIO nanoparticles have recently been demonstrated for biocompatibility in both preclinical and clinical trials [2,16,17,18,19,20]. Uniquely, SPIO nanoparticles have also been investigated as a medical device for localised hyperthermia therapies to induce cell death in cancer tissues [21,22].

By applying an alternating magnetic field, SPIO nanoparticles rotate, and their kinetic energy is translated into a localised increase in tissue temperature (up to ca. 46 °C), resulting in targeted cellular apoptosis with minimal side effects [23,24]. SPIO nanoparticles in the range of 20–150 nm were reported to have good renal clearance and avoid opsonisation with serum proteins, before being removed from the bloodstream by macrophages; this can be significantly quickened by controlling the surface charge on the SPIO via the addition of a surface coating [16,17]. Functionalising SPIO can also prevent their degradation and convert them into efficient carriers for pH-triggered drug release [25], as well as suppressing their aggregation [18,26]. These surface coatings can also be utilised to attach prodrugs; thus, the whole functional nano-complex has both diagnostic and therapeutic potential properties [27]. However, translating metal oxide nanoparticle formulations from the aqueous solutions into physiological conditions is problematic, due to their immediate or gradual sedimentation in ion-rich media conditions, such as physiological buffers and bodily fluids [25], which inherently results in protein binding and the flocculation of SPIO nanoparticles.

Carnosine is a naturally occurring dipeptide, consisting of β-alanine and L-histidine, which is generally located in the brain and muscle tissue [28]. Its positive effect on cell cycle regulation, from DNA synthesis to chromosome condensation, has led scientists to investigate carnosine as a potential therapeutic in anti-ageing, immune-stimulation and wound healing therapies [29]. More recently, its natural oxidative metabolic regulation in both muscle and brain tissues has also been demonstrated [30,31,32]. Carnosine’s anti-cancer properties have been investigated using in vitro models [33]; specifically, it has been investigated as an anti-human glioblastoma multiforme [34]. However, the physiological levels of carnosine are inherently low, and are below the threshold required to see any of the significant effects reported above [28,35]. Due to the rapid enzymatic metabolism of carnosine in the body, it is imperative that any therapeutic application protects against this. Herein, we describe the synthesis of superparamagnetic iron oxide nano-rods (IONRs) coated with a branched polymer that can be loaded and unloaded with carnosine, using mild hyperthermia (such as that produced from oscillating IONR in a magnetic field) as a controlled release carrier that is stable under physiological conditions. The potential therapeutic properties of this new vector are demonstrated in vitro using the tumour cell line U87 MG, a human primary glioblastoma cell line. The superparamagnetic properties of this carrier have been utilised as an MRI contrast agent towards a targeted glioblastoma treatment.

## 2. Materials and Methods

### 2.1. Synthesis of Iron Oxide Nanorods

All reagents were purchased from Sigma-Aldrich (Dorset, UK), were of analytical grade and used without further purification. Aqueous polyethyleneimine (PEI) (1 mL, 3000–6000 Mw, 50% aq) was dissolved in distilled water (80 mL) and transferred into a PTFE lined 4748 Parr acid digestion vessel. Iron (II) sulfate heptahydrate (0.556 g, 2.0 mmol) and anhydrous iron (III) chloride (0.649 g, 4 mmol) were then added to the polymer solution, followed by cyclohexane (16 mL). The reaction was gently stirred at room temperature until the iron salts were fully dissolved, with the cyclohexane layer helping to prevent unwanted oxidation. The stirring rate was then increased (900 rpm) and ethylenediamine (12 mL) immediately added before sealing of the reaction vessel. The reaction was then heated (120 °C) with stirring (900 rpm) for 20 h. On cooling and pressure equalisations, the magnetite product was separated using a neodymium magnet (0.5 tesla) and washed with dry dimethyl formamide (3 × 20 mL), before being stored as a suspension in dimethyl formamide.

### 2.2. Surface Coating towards Water Dispersible Colloid

The suspended iron oxide nano-rods (120 mg/10 mL) were transferred to a dry double-necked round bottom flask before the addition of branched polyethyleneimine (BPEI) (1.2 mL). The reaction mixture was stirred (1000 rpm) and heated (70 °C) overnight under nitrogen atmosphere. The final product was separated using a neodymium magnet (0.5 tesla) and washed with DMF (3 × 30 mL) thoroughly to remove the free PEI. The polymer coating was then protonated via the addition of HCl (6 M, 0.3 mL), resulting in an overall pH of 3. The coated nano-rods were then separated via ultra-centrifugation (15 min, 15,000 RPM) and stored under nitrogen as a dry powder. The synthesis yield was 84.4 ± 5.5%, with a coating ratio of 13.4 ± 2.3% surrounding IONRs of 85.9 ± 17.1 nm length. The dimension measurements are available in the Appendix A.

### 2.3. The Characterisation of IONRs

#### 2.3.1. General Information

The morphology of PEI-coated Fe_3_O_4_ nano-rods was characterised by scanning electron microscopy/energy dispersive X-ray spectroscopy (SEM-EDS, JSM-7100f, JEOL Ltd., Tokyo, Japan) integrated with Aztec software (version 2.4, Oxford Instruments, Oxford, UK). In addition, the dimensions were obtained by transmission electronic microscopy (TEM, JEM 2100, JEOL Ltd., Tokyo, Japan) at 200 kV followed by analysis using ImageJ software (version 1.44, National Institute of Mental Health, Bethesda, MD, USA). The crystalline structure of the coated IONRs was measured by powder X-ray diffraction (X’Pert PRO, PANalytical) generating Cu Kα radiation at a wavelength of 1.5406 Å and analysed by high score software (version 5.1 Malvern Panalytical. Ltd., Malvern, UK). Infra-red spectra were recorded using an FT-IR spectrometer (PerkinElmer. Ltd., Waltham, MA, USA). The thermogravimetric analysis was carried out using a thermogravimetric analyser (TGA 4000, (PerkinElmer. Ltd., Waltham, MA, USA) in tandem with a GC-MS system (GC Clarus 580-MS Clarus SQ 8S, (PerkinElmer. Ltd., Waltham, MA, USA). To study the suspension stability profile, NanoPlus (Particulate systems, Norcross, GA, USA) was used to measure the dynamic light scattering (DLS) of the hydrodynamic diameter size and zeta potential for the surface charge. The magnetic properties of the IONRs were recorded by a 1.5 T whole body MRI system (Avanti, Siemens, Munich, Germany) applied on a 2 mm thickness phantom.

#### 2.3.2. SEM-EDS

Scanning Electron Microscopy and Energy Dispersive X-Ray Spectroscopy were integrated to use electrons for imaging while the X-rays were used for characteristic chemical information on the iron oxide nano-rods. SEM-EDS was used to identify the dried BPEI-coated IONRs on a copper grid or the fixed cells on a plastic slip. Each sample was attached to a carbon tape on an aluminium stub (13 mm), then sputter coated with 5 nm conductive material (gold). Appendix A show that EDX spectra were processed to localise the nano-rods and analyse different elements such as carbon, oxygen, and iron ions using Aztec software (Oxford Instruments, Abingdon, UK).

#### 2.3.3. TEM

TEM was used to determine the morphology and nano-rods dimensions. One drop of the diluted BPEI-coated IONRs’ colloid was placed on a Formvar carbon coated copper grid (300 mesh), then left to air dry and visualised using TEM at 200 kV. The average dimensions with the standard deviation were determined by measuring the length, width, depth, and hypotenuse of rods from many images.

#### 2.3.4. FT-IR

Fourier transform infrared spectra were acquired by FT-IR spectrum on the dry BPEI-coated IONR powder after checking a blank background.

#### 2.3.5. XRD

The crystalline structure of the iron oxide nano-rods powder was analysed by high score software after obtaining the diffractograms using an x-ray diffractometer (X’Pert PRO, PANalytical, Malvern, UK). The device provided a Cu Kα radiation source (λ = 1.5406 Å) and scan at 30 mA and 40 kV in slow mode over a range of (10–70, 2θ).

#### 2.3.6. TGA-GC-MS

The TGA was run in series with a GC-MS system. The dry BPEI-coated IONR powder, either bare or coated with PEI, was weighed in a platinum crucible after subtracting the crucible weight. Gradual temperature was applied starting from 50 to 800 °C. The loss of weight reflected the identification and quantification of the coating material (Appendix A). Carnosine alone or as a final surface coating was characterised using the same method. The carnosine was loaded into the capped IONR in the final step of the preparation, by adding an aqueous solution of carnosine (ca. 0.25 mL, 1 mol) to a freshly prepared solution of the carrier (50 µg/mL) and rapidly stirring using a vortex mixer (ca. 1 min). Excess carnosine was removed by centrifugation and pellet dried under vacuum, resulting in a black powder loaded with carnosine. In Appendix A, the peaks of the produced gas were separated by GC and matched with MS library to identify the organic compounds.

#### 2.3.7. DLS

The DLS was used to measure the hydrodynamic diameter and zeta potential of the surface of the BPEI-coated IONRs. The suspension, with final concertation of 100 µg/mL in distilled water, was prepared in neutral distilled water and acidic pH 3 conditions by adding a drop of HCl 1M. Each measurement consisted of 60 accumulations, repeated in triplicate and then averaged to minimise the error. For surface charge measurement, the zeta quartz cell was filled with the diluted suspension and, after checking the optimum density for starting the test, the zeta potential was obtained in triplicate.

#### 2.3.8. MRI Analysis

MRI measurements were performed at room temperature and used to measure T2 relaxation, to investigate the correlation between the relaxivity and sedimentation of the BPEI-coated IONRs. The colloidal stability, with respect to sedimentation, was observed via MRI in physiological solutions: 4-(2-hydroxyethyl)-1-piperazineethanesulfonic acid (HEPES), phosphate-buffered saline (PBS), fetal calf serum (FCS), Eagle’s minimal essential medium (MEM), Improved Minimal Essential Medium/Reduced-Serum (Opti-MEM^®^), Gibco™, Thermo Fisher Scientific, Waltham, MA, USA), and with distilled water at both pH 3 and 7. The IONR (50 µg/mL) were suspended into HEPES, PBS, FCS, MEM, Opti-MEM, distilled water (pH 3 and pH 7) and FCS combined with Opti-MEM (1:10 *v*/*v*) (200 µL) and transferred into a flat-bottom 96 well plate (with 10.67 mm depth × 6.86 mm top diameter × 6.35 mm bottom diameter wells). Images were collected at one-hour intervals over two days using a 1.5 T MRI scanner (Avanto, Siemens, Munich, Germany) with transverse slices centred on the 8 rows of wells in the 96 well plate. Images were acquired using a proton density weighted Turbo Spin Echo sequence (TR = 7.14 s and TE = 34 ms) with a 512 × 160 acquisition matrix and an isotropic resolution of 350 μm. An initial image was taken as a reference and was subtracted from the images collected at 1 h intervals to reveal any sedimentation. Where no sedimentation occurred, the resulting value was be zero (corresponding to dark blue) whereas any sedimentation (occurring after the reference image was collected) resulted in an increase in signal (corresponding to yellow through green coloured image).

### 2.4. Cell Study

#### 2.4.1. Cell Culture

The human glioblastoma U87 MG-Red-FLuc (Bioware Brite, PerkinElmer, Waltham, MA, USA) was the authorised cell line in all bio experiments. These cells were incubated in Opti-MEM Reduced Serum Medium (Gibco™, Thermo Fisher Scientific, Waltham, MA, USA) culture medium, supplemented with fatal bovine serum up to 10%. The antibiotic puromycin (Gibco^®^, Thermo Fisher Scientific Waltham, MA, USA) was added after the initial thaw at 2 µg/mL. The incubation was at 37 °C in a humidified atmosphere containing 5% CO_2_.

#### 2.4.2. Cell Viability Assay

The cell viabilities were assessed by MTT assays, wherein, U87 cells were seeded into 96-well plates at a density of 4000 cells per well in 200 mL of growth media. The cells were left for adherence (18 h) before treatment. Media were then gently aspirated and replaced with different concentrations of carnosine. The BPEI-coated IONRs were sterilised by filtration through a 0.22 μm syringe filter, and concentrations of iron oxide (0, 1, 5, 10 µg/mL) were tested in three replicates using three passages. After treatment durations of 48 h, 20 µL of the MTT assay solution 3-(4,5-dimethylthiazol-2-yl)-2,5- diphenyltetrazolium bromide (0.5 mg/mL in PBS Sigma Aldrich) were added to each well. Cells were subsequently incubated (37 °C for 2 h) and the media solutions was gently aspirated before being replaced with dimethyl sulfoxide (100 mL). The plate was wrapped with tin foil and left on the shaker for 10 min. The plate reader (Clariostar, BMG LABTECH, Ortenberg, Germany) was used to read the absorbance at 570 nm.

#### 2.4.3. Live Cell Imaging (IncuCyte System)

##### Proliferation, Cytotoxicity

All studies were observed on U87 cells using the real-time live cell analysis after applying the treatment of different carnosine concentrations. For proliferation and cytotoxicity experiments, cells were seeded in a 96-well plate at a density of 4000 cells per well and left for 18 h for adherence. Concentrations of carnosine (0, 10, 15, 25, 100 mM) were added in the presence of IncuCyte Cytotox red for counting dead cells (250 nM, Essen Bioscience). The cyanine nucleic acid dye permeated cells with compromised cell membranes. Using the IncuCyte S3 Live Cell Analysis System (Essen Bioscience Inc., Ann Arbor, MI, USA), images were snapped with a 10× objective lenses in each well every three hours whilst remaining inside the incubator (48 h).

##### Migration and Invasion

The ratio of the proliferation phase, and percentages of the red dye and cytotoxicity levels, were calculated using the integrated Incucyte^®^ standard software (version 2020B, Essen Bioscience Inc., Ann Arbor, MI, USA). For the migration and invasion tests, cells were seeded in a 96-well plate at a density of 10^4^ cells per well and left for 18 h for adherence. Carnosine (0, 50, 100, 150, 200 mM) was added following the optimised protocol from the IncuCyte scratch wound assay (Essen BioScience, Sartorius, Göttingen, Germany). The base matrix of Matrigel, for 3D culture, was prepared as reported in the Corning protocol guidelines. The migration test was subsequently repeated with the addition of mitomycin C (10 μg/mL, Sigma-Aldrich, St. Louis, MO, USA) after cell seeding and settling, one hour before the final carnosine treatment. Validation experiments were conducted to assess the effect of mitomycin C on the interaction with carnosine using proliferation IncuCyte phase images, the viability of the cells was assessed via a Trypan blue exclusion test, and adherence was assessed by comparative cell counting. The percentages of relative wound density and wound width were calculated using the integrated IncuCyte S3 wound scratch software (version 2020B, Essen Bioscience Inc., Ann Arbor, MI, USA).

#### 2.4.4. Uptake and Localisation (SEM, TEM)

U87 cells were grown at a density of 4000 cell/well on cover slips using 6-well plates, then treated with BPEI-coated IONRs (5 μm/mL) or carnosine (25 µM), and incubated (24 h at 37 °C and 5% CO2). The media were aspirated, and the cell monolayer was washed with HEPES buffer followed by the addition of 4% paraformaldehyde for 10 min. Cells were then washed three times with HEPES and dehydrated in graded alcohol solutions of (50%, 60%, 70%, 80%, 90% and 100% ethanol) for five minutes each. The slips were then coated with gold (5 nm) using a sputter coater (Q150R ES, Quorum, Madrid, Spain), then imaged using SEM (probe current of10 mA and accelerating voltage of 10 kV). Using energy-dispersive X-ray spectroscopy (EDS), the spectrum was collected using Aztec software.

For TEM imaging, cell fixation steps were applied. Double strength fixative (6% glutaraldehyde in 0.2 M phosphate buffer, pH 7.4) was added to complete growth media at ratio of 1:1. The previously rinsed cells in the 96-well plate were subjected to glutaraldehyde-media solution at normal growth conditions (37 °C, 5% CO_2_). After 1 h, the cells were gently washed with phosphate buffer (1 mL, 0.2 M at pH 7.4) three times. After 1 and 4 h, the fixed cell monolayers were aspirated cautiously, then transferred to 5 mL centrifuge tubes and covered with excess phosphate buffer to during sedimentation at 4 °C overnight. The resin was prepared by mixing Araldite CY212 resin (25 mL), Agar 100 resin (15 mL) and DDAS (55 mL). Then, of dibutyl phthalate (2 mL) and of DMP 30 (1.5 mL) were blended with the resin mix. The intensive stirring continued until no striations were observed in the final combination. Finally, the cells were embedded with a mixture of resin: propylene oxide at 1:3 *v*/*v* ratio for 2 h, followed by overnight at a 1:1 *v*/*v* ratio. On the next day, the resin was completely replaced with fresh resin in new vials and left for 3 h. Meanwhile, the cells were dehydrated via a series of ethanol treatments (50%, 70% and 90% and 100% ethanol) for 10 min. Osmium tetroxide was added to produce excellent fixation of the cytoplasm, then neat propylene oxide was used twice for 10 min. Finally, the resin was added to the cell to polymerise through exposure to a temperature of 60 °C for 3 days. Resin blocks were sectioned using a microtome with a diamond knife attached to a boat, which was filled with DI water. The thickness of each slice was adjusted to 70 nm. Each section was mounted on Formvar films on a copper TEM grid (200 mesh) and left for air drying. The images were obtained by TEM (accelerating voltage of 80 kV).

#### 2.4.5. The Controlled Release of (NRs/Carnosine) Synergism

##### Proliferation, Cytotoxicity (MTT)

The MTT assays were utilised as an index of cell viability. The U87 cells were seeded and left to adhere for 18 h to test the existing cell viability. Media were then gently aspirated and replaced with carnosine (25 μM). The BPEI-coated IONR controls (2.5, 5 µM) were prepared. Two sets of treatments were prepared by separately adding each of the BPEI-coated IONRs for a row of (media/carnosine) wells and repeating the same steps instantaneously in three plates. A plate was incubated for 3 h at 40 °C and another plate for 5 h at 40 °C, then each was moved to the normal growth conditions (37 °C, 5% CO_2_), where a third plate was also incubated. After overall treatment durations of 48 h, the MTT assays were completed to obtain the absorbance. A similar test was applied on different plates after changing the order of treatment addition in the final two mixtures by vortexing them for 1 min before adding the mixture to the media. Premixing allows the same concentration of carnosine to be embedded in the branches of two PEI concentrations.

### 2.5. Dialysis Membrane Tubing Test/Liquid Chromatography Mass Spectrum (LC-MS) Assay

The sustained release simulation for carnosine was mimicked by applying the dialysis membrane experiment. Dialysis tubing contains pores that were 1–10 nm in diameter, and it is semi-permeable. A dialysis tubing was filled with different types of carnosine, as well as free or loaded coated IONRs, and suspended in a beaker of water under controlled conditions. Samples were collected each hour and the assay was performed using a SCIEX TripleTOF 5600+ (AB Sciex LLC, Framingham, MA, USA). Positive ion mass spectra were acquired via direct infusion for optimisation and a hyphenated LC system. For quantification, we prepared a standard curve of carnosine at a range of concentrations (1.0, 2.0, 5.0, 10.0, 15.0 μM). The standards and samples, mixed with water, were introduced into the source at a flow rate of 5 μL min^−1^. The LC conditions were as follows: The separation was performed on an Eksigent ekspert nanoLC 425 system ACE AQ column (0.5 × 150 mm^2^, 3 μm) using ACE chromatography. The mobile phase was composed of MS grade water with 0.1% formic acid (solvent A) and acetonitrile with 0.1% formic acid (solvent B) (Merck, UK). The standardised gradient elution program in LC separation was: 0 min, 1% B; 1 min, 1% B; 3 min, 50% B; 6 min, 90% B; 9 min, 90% B; 10 min 1% B re-equilibrate for 3 min. To identify the transitions of carnosine, full scan MS/MS was performed by fragmenting the precursor ion (227.11 *m*/*z*) of carnosine at a fixed collision energy of 25 V and a declustering potential of 80 V. The transitions were acquired in the mass range of 100 to 300 *m*/*z*. The column temperature was controlled at 45 °C and the sample chamber temperature at 8 °C, with an injection volume of 2.0 μL. The ion spray voltage was fixed at 5.5 kV, Gas 1 = 12, Gas 2 = 0, using a Duospray source and 50 μm electrode. Data were analysed using Multiquant 3.0.3 (SCIEX, AB Sciex LLC, Framingham, MA, USA).

### 2.6. Statistical Analysis

The data were presented as the mean ± the standard error of the mean (SEM). For the statistical analyses, a Student’s *t*-test (unpaired samples) was applied to compare groups after applying different treatments. The one-way analysis of variance (ANOVA) was carried out for multiple comparisons between the control and each group using Dunnett’s post-test. All graphs and analysis were demonstrated by GraphPad Prism (version. 8 software, Inc., La Jolla, CA, USA). The value 0.05 was selected as the statistical significance level and indicated with (*) for *p* < 0.05, (**) for *p* < 0.01, (***) for *p* < 0.001 and (****) for *p* < 0.0001.

## 3. Results and Discussion

### 3.1. Synthesis and Characterisation

The synthetic route to embed carnosine within an IONR-functionalised carrier is illustrated in (Figure 1). Branched polyethyleneimine polymer (BPEI), with its high density of cationic amino functional groups, dendritic structure, and inherent interstitial voids, was selected as a soft template for the IONRs [15]. In step one of the synthesis, the ferrous and ferric iron precursors were loaded into the polymer template before the addition of ethylenediamine, resulting in the co-precipitation of Fe_3_O_4_. The template was then removed, in step 2, via thermal degradation in a sealed reaction vessel (120 °C, 20 h), and the IONRs were washed with dry dimethyl formamide to avoid aggregation. TEM, FTIR and powder X-ray diffraction (XRD) studies confirmed that the IONRs were (85.9 ± 17.1 × 19.1 ± 1.8 × 11.3 ± 2.1 nm) Fe_3_O_4_ nanoscale crystals (Figure 1 and Appendix A).

The IONRs were then capped in fresh BPEI (13.4 ± 2.3% *w*/*w* coating ratio), an efficient transfection agent [36], before being washed with water, centrifuged, and dried under vacuum, resulting in a black powder (84.4 ± 5.5% yield), before being stored under a nitrogen atmosphere until further use [37,38,39,40,41,42,43]. The powder XRD spectrum of the capped IONRs corresponded [16], as expected, to the two individual components, with the inorganic lattice displaying the (111, 220, 311, 222, 400, 511, 440) planes of Fe_3_O_4_ (Appendix A). Thermal gravimetric analysis (TGA), with hyphenated FTIR-GC-MS (shown in Appendix A), confirmed that the IONRs were coated with BPEI (87% *w*/*w*) [1,37]. TEM microscopy (Figure 1B) of the BPEI-coated IONRs was found to be crystalline, with a tetragonal prism-dipyramid morphology and mean length of 85.9 ± 17.1 nm. The electron microscopy images show prolific agglomeration and a lack of uniformity prior to coating with the BPEI, whereupon they transition to monodispersed tetragonal bipyramidal nano-rods (Figure 1 and Appendix A) [37,44]. The BPEI polymer adsorbs onto the surface of the BPEI-coated IONRs, thus making a steric stabilisation, which enhances the dispersion state. Optimising the polymer density on the surface of the BPEI-coated IONRs is essential to avoid the cross-bridging effects reported at higher concentrations [45].

Figure 2 shows a carnosine loading of ca. 60% mg carnosine/µg IONR/mL water, as analysed by evolved gas analysis using TGA-FTIR-GC-MS. The polymer is observed, in Figure 2, to be completely dissociated between 150 and 300 °C, while the β-alanine and L-histidine from the carnosine degraded between 300 and 700 °C. This is similar to the results observed for the pure carnosine and the two amino acids (β-alanine between 200 and 300 °C and L-histidine between 600 and 900 °C (Appendix A), evidencing that carnosine is embedded within the structure of the carrier [46].

The FT-IR attenuated total reflectance (ATR) spectrum of the carnosine loaded carrier displays characteristic bending bands between 1474 and 1630 cm^−1^ that are attributed to the peptide bond of carnosine, and a band at 2900 cm^−1^ from the imidazole ring of L-histidine. In addition, the strong stretch at 3000 cm^−1^ is due to the absorbance due to alkane C-H (Figure 3) [37], and the peptide bond overlaps the strong bending band at 1474 and 1630 cm^−1^ of the NH2 group. Despite the annealing of the BPEI as a template during the synthesis of the IONRs, it is clear that there remains trace residual polymer on the nanorods.

### 3.2. Colloidal Stability

Aggregation of nanoparticles can lead to anomalies in results and limit reproducibility. As a time-dependent process, this can result in changes when measuring their cellular responses and the toxicity profiles [39,45,47]. Metal oxide nanoparticles, including those investigated as MRI contrast agents, are often susceptible to agglomeration or sedimentation over prolonged periods of time, especially under physiological conditions [10,43]. Therefore, the colloidal stability of SPIONs is commonly demonstrated in deionized water in order to eliminate the detrimental interactions of electrolytes and variations in pH. This is, however, not representative of real-world behaviour and, thus, their colloidal stability in physiological solutions and media is of paramount importance when considering their potential biomedical applications [47,48]. The zeta potential is often used as an indication of the net charge of a cloud of counter ions around the particle, which differ because of their environment due to the ionization across the surface groups [45,49,50]. It is reported that the addition of a BPEI coating, onto metal oxide nanoparticles, improves charge and overall electro-steric stabilisation via the induced steric hindrance provided by the formation of a hydrophilic macro-shield [39].

In order to evaluate the stability of the BPEI-coated IONRs, DLS measurements are usually conducted [51]. However, due to the limitation in DLS sensitivity when using biological buffers, serum, and complex media, they often lead to inaccurate refractive index results [52]. We therefore studied the samples, described herein, using biological media and MRI techniques. Thanks to the paramagnetic properties of the IONRs, MRI can be utilised to provide spatial resolution and contrast of the samples. The stability of the IONR in water was evaluated for hydrodynamic diameter and polydispersity changes, measured using dynamic light scattering (DLS) over the course of one hour and 30 days (Figure 4).

The IONR size and polydispersity did not show any significant changes over time in water at pH 7, indicating that the formulation is colloidally stable for over one hour at room temperature (Appendix A). By comparison, the uncoated IONRs show significant aggregation even after sonication, with a mean hydrodynamic diameter of ca. 500 nm (Figure 4A). Reducing the pH down to three did not have a significant effect on the hydrodynamic diameter of the IONR; however, the polydispersity was observed to be more uniform (Figure 4). The R^2^ of the coated IONRs, at both pH 3 and 7, was below 0.5 and, therefore, has an insignificant effect on the hydrodynamic diameter and polydispersity. However, the polydispersity for the uncoated IONRs was 0.65, which refers to a moderate effect in the uniformity from the first hour (Appendix A). The initial hydrodynamic diameter and polydispersity were ca. 165, 0.2), and ended ca. 500, 0.35 after one hour.

The results of the 30 day DLS stability trial show that at pH 7, the IONR has stability in the hydrodynamic diameter (172 ± 8.8 nm). Similar results were also observed at pH 3, with a mean hydrodynamic diameter of 163 ± 8 nm. There was an absence of aggregation indications and no precipitation would have been developed over a month, which is essential for biomedical applications (Figure 4). [38]. This study showed a wide range of safe pH storage conditions as a ready suspension that will be buffered before the application instead of a longer process of desolating the dry powder by sonication. Indeed, the suspension had been reserved over a year and did not show sedimentation or change in specifications.

In line with the DLS results, the two dispersions of IONR in water, at pH 3 and 7, show uniform dispersion throughout the 48 h, with little to no evidence of any sedimentation or flocculation of the carrier. The carriers quickly agglomerated with the addition of PBS, resulting in a black precipitate that completely fell out of solution within 48 h, yielding similar results for the 1 h and 48 h images. PBS (1×) showed rapid sedimentation, while sedimentation of the carrier was slower over the course of the experiment when dispersed into HEPES buffer. The main ingredient of the Eagle’s minimal essential medium (MEM) is PBS and, therefore, it was substituted with OptiMEM media, which contains HEPES. The carrier was observed to remain colloidal in the third component of the cell media, FCS; however, when combined with Eagle’s minimal essential medium (MEM), the IONR was sedimented out of the solution. Conversely, when Opti-MEM-reduced serum media was substituted for MEM, the IONR remained suspended as a colloidal dispersion (Figure 5).

The MRI results included herein confirm that the use of PBS, alone or within a media, results in rapid sedimentation. This is due to the relatively high sodium ion concentrations and the subsequent compression of the electrical double layer that surrounds the BPEI-coated IONRs [52], and ultimately the collapse of the colloidal system. It is also reported that sedimentation can be reduced by substituting PBS for the zwitterionic HEPES buffer since it is proposed that the FCS proteins attach to the surface of the BPEI-coated IONRs and sterically hinder any potential to agglomerate [45]. This further MRI experiment was undertaken simultaneously to evaluate this process by determining the effective spin lattice relaxation time constant (T2^eff^) since the paramagnetic properties of the nanorods are known to shorten this value [53]. Our results show that the ionic charge, between the BPEI-coated IONRs and media, can be balanced by using Opti-MEM as the media in combination with FSC. Furthermore, the BPEI-coated IONRs within the carrier preserve their super paramagnetic properties, enabling them to be utilised as MRI contrast agents. The overall nano-carrier foundation is also proven to be stable as both a dry powder—that can be readily re-suspended—and directly as a suspension in sterilised water, for at least one month.

### 3.3. Proliferation and Cytotoxicity (IC_50_/EC_50_)

Prior to the proliferation and cytotoxic studies of the U87 MG cell line, cell density studies were conducted to define the cell number required to attain significant adherence to the bottom of the 96 well plate within 18 h and without reaching full confluence over 48 h. As such, the optimum cell density was found to be 4000 cells per well. The inhibition in cell proliferation upon increasing the carnosine concentration from 0 to 125 mM after 18 h, to these seeded cell cultures, can be seen in Figure 6A,B and Appendix A.

The U87 cells were imaged hourly during the 48 h exposure in complete culture media. The growth of U87 was relatively inhibited after 8 h and completely affected after 48 h when the applied carnosine reached ca. 125 mM. To confirm that this was not due to the cells being compromised in these wells, U87 cells were first seeded, and their proliferation was monitored for 24 h, before carnosine was added. Figure 6B clearly shows the suppression of U87 growth within 8 h after the addition of carnosine, which led to a complete growth inhibition and cell death after 2 days (Figure 6D). The half maximal inhibitory concentration (IC_50_) was determined after the MTT assay (a colorimetric assay for assessing cell metabolic activity) was performed, and was analysed using GraphPad Prism software (San Diego, CA, USA). A range of (23.6 to 34.7 mM) carnosine with a confidence level of 95% and a regression value of R^2^ = 0.8195 was found (Figure 6C). The effect of carnosine on U87 was clearly observed using phase contrast images of the cells with and without carnosine, using the IncuCyte^®^ CytoLight Rapid green dye, while IncuCyte^®^ Cytotox red reagent was used for real-time quantification of cell death. The number of live cells was shown to decrease while the number of dying cells increased in a dose-dependent manner.

The analysis of the phase images and red channel enabled real-time evaluation of cell membrane integrity and cell death in response to carnosine exposure after 24 and 48 h (Figure 7). The maximal effective concentration curve was defined (EC_50_ = 28.2 mM).

The red fluorescent areas with blue masks were quantified for different carnosine treatments using IncuCyte^®^ standard analysis software. This was integrated with the previous proliferation test at (IC_50_ = 34 mM) (Figure 8). The plots representing live and dead cells initially showed no effect, followed by a cytostatic phase starting from 30 mM carnosine (Figure 8). However, the cell cytotoxicity could be seen to start immediately after the integration point. The mechanism of cell death is the effect of carnosine on the functionality of mitochondria, which lose the metabolic capability to develop the purple colour in the MTT assay and allow the red dye to penetrate. That accounts for the similarity of IC_50_/EC_50_ in the MTT, proliferation and cytotoxicity tests.

### 3.4. Cell Migration and Invasion

In vitro migration was studied by creating a wound in a cell monolayer, then capturing the images at regular intervals during the closure of the wound due to the migration of the cells. By comparing the images of the same cells in media alone, it is possible to quantify the migration rate of the cells, which reflects the effect of the treatment on cell mobility [54,55]. The invasion assay, however, is studied by embedding a monolayer of tumour cells inside a biomatrix to monitor the speed at which the cells invade through the matrix to the neighbour cells [55,56,57]. To study the effect of carnosine on metastasis, migration and invasion were assessed over a 48 h course in the same plate in the presence of a concentration of mitomycin C that is known to prevent cell proliferation. The quantification of phase images assesses cell morphology at every time point. After proliferation was suppressed by mitomycin C, migration was reduced after 24 h but not abolished by carnosine (*p* < 0.2 from 100 mM, 6 experimental replicates). The migration showed a plateau in wound density over the second day because mitomycin C formed covalent cross links between complementary strands of DNA, thereby preventing their separation and inhibiting DNA replication (Figure 9) [58,59]. However, proliferation and migration were also observed to decrease following the addition of carnosine. In a time-dependent manner, both effects were seen to increase after 24 h, with an effective decrease in concentration of intracellular ATP (Appendix A) [34].

The data were used as indicators to recalculate the IC_50_/EC_50_. The IC_50_ was determined by the relative wound density ratio and the EC_50_ was determined using the width of the wound, and these values corroborated the results found from live/dead cell curves. The IC_50_/EC_50_ are ca. 100 mM at a density of 10^3^ cell/well (Figure 10). To ensure the independency and non-toxicity of the effect of mitomycin C on carnosine, a series of tests validated the cell proliferation, viability, and adhesion (Appendix A) [60,61]. No synergism appeared between carnosine and mitomycin C, which was proved by imaging of live cells during their migration, conducted to monitor intracellular interactions [54]. In addition, the viability and adherence were similar.

Carnosine affects migration and metastasis by suppressing the cells’ mobility regardless of the effect of proliferation [61]. The effects of the cell matrix and interactions in cell invasion mimic the cells during wound healing in vivo. Carnosine affects the invasions and metastases significantly when the concentration exceeds 50 mM (Figure 11).

### 3.5. Cell Viability with IONRs (MTT)

The quantitative MTT assay data shows that the viability of cells treated with BPEI-coated IONRs at concentrations of (0–10 µg/mL) did not display a significant statistical difference in one passage. However, two other passages showed cytotoxicity caused by IONRs at a concentration of 10 µg/mL. Hence, the surface modifications, which occur via positive charge (+19.14 mV) at pH 7, are safe to be applied on U87 cells using BPEI-coated IONRs concentrations of less than 10 µg/mL for an exposure period of 48 h (Figure 12). The effect of BPEI-coated IONRs is nonselective, and an MTT assay was applied for comparison. U87MG cells were grown in serum-free medium supplemented with dibutyryl cyclic adenosine monophosphate (dbcAMP; 0.3 mM) and B27 Supplement (1% *v*/*v*). The IONRs showed less effect on the metabolism of the differentiated U87 cells [62]. However, it was not a significant difference (Appendix A). In fact, astrocytes harbour protective mechanisms including neurotrophic factors and anti-oxidative stress molecules [63,64].

### 3.6. Uptake and Localisation of (NRs/Carnosine) in U87 Cells

The intercellular transportation and intracellular localisation of BPEI-coated IONRs are vital in their overall interactions due to their theragnostic effects [65]. Many factors impact the uptake of IONR, such as particle size, polydispersity, and the surface charge of the functionalised coating. The SEM images show the uptake and localisation of BPEI-coated IONRs in U87 cells after incubation for 24 h. A remarkable number of nano-rods were spread in the cytoplasm, mainly localised inside the nucleus. The iron ions overloaded heavily in the images of cells treated with BPEI-coated IONRs at a concentration of 10 μg/mL; however, a concentration of 5 μg/mL kept the cells alive. The energy-dispersive X-ray spectroscopy (EDS) detected the elements and proved the iron peaks in the map scan of the cells. This toxicity was dictated in previous works when the BPEI-coated IONRs became overloaded ca. 10 μg/mL [38], while the BPEI-coated IONRs at a concentration of around 5 μg/mL can be considered safe for use in the treatment of glioblastoma brain tumours [1] (Figure 13).

Nanomaterial with positively charged surface functionalisation allows cellular adhesion and transportation through the ionic interaction with negatively charged cell surfaces, which enhances their cytotoxicity margin [66,67,68]. The BPEI-coated IONRs may cause DNA damage by breaking the hydrogen bonding, which generates the reactive oxygen species (ROS) and, at high doses, inside the nucleus, resulting in a structurally compromised actin cytoskeleton structure [64] (Figure 14). TEM images were taken of U87 cell sections to support the entrance of the rods inside the cells, as opposed to merely settling on the surface. The blank cell shows the basic component of the cell. The sections of the cells, after one hour of incubation with BPEI-coated IONRs at a concentration of 5 µg/mL, exhibited BPEI-coated IONRs that were spread around the cell membrane. Time-lapsed electron microscopy images (Figure 15A–I and Appendix A) of the tumour cells, after 4 h of exposure to the BPEI-coated IONRs, demonstrated the capability of rods <100 nm to penetrate the cell border and reach the cytoplasm that was moving towards the nucleus regions. The sections from the U87 cells after 4 h of carnosine treatment demonstrated changes in the cell morphology as the cells shrank into a spherical shape. In addition, the free carnosine accumulate inside the mitochondria as a dark silhouette. The uptake of carnosine is likely to have taken place during mitosis, whereupon the carnosine was transferred into the daughter cells; however, the full chemical effect of the carnosine was not observed until later in the cell cycle (Figure 15J–M).

The SEM-EDS and TEM images enabled the tracking of entry into U87 cells over (1, 4, 24) hours, and the subcellular release and distribution of the drug that was related to the accumulation of BPEI-coated IONRs in the target organelle nucleus [64,69]. The TEM images represent the autophagy vacuoles in the U87 cells’ cytoplasm after 4 h of carnosine treatment, which was similar to that of rapamycin [70]. The free carnosine increase the damage caused to the structure of the mitochondria, which was previously reflected in the MTT test.

### 3.7. The Effect of (IONRs/Carnosine) Synergism in Controlled Release

Using dialysis tubing to model the release of carnosine, Figure 16 predicts that water entered via osmosis, and the carnosine left the tubing via diffusion. The IONRs’ BPEI-coated size prevented them from having permeability through the membrane. Increasing the temperature to 40 °C accelerated the complete release of carnosine after 2 h. The BPEI-coated IONRs partially retained the carnosine at 37 °C. Free carnosine had a linear trend of release, which might have extended until 8 h, thus delaying the effect of the treatment, as mentioned in the proliferation study. Using the carnosine loaded BPEI-coated IONRs may offer the advantage of a more quickly controlled treatment relative to the traditional application of carnosine. By comparing the MS spectra of the initial and final samples, it can be seen that the carnosine structure was not affected by the mild hyperthermia. All samples and standards showed identical peaks, which appeared due to the release of carnosine without any impurities or structural modification (Appendix A).

In all plates, the controls of seeded cells, alone or with carnosine, showed no significant changes in terms of live cells, which proves the lack of an effect caused by mild hyperthermia conditions (40 °C, 5% CO2) on cell growth. In the samples of bonded IONR nano-carrier with 25 mM carnosine, the normal incubation in (37 °C, 5% CO2) showed that the effect of carnosine was hindered because the polymer it attached to the BPEI-coated IONRs. However, applying heat of 40 °C for 3 h initiated the release of carnosine.

A maximum of 5 h of mild hyperthermia could de-attach the carnosine inside the cells, which was reflected in the treatment result after the completion of 48 h in normal incubation (Figure 17A). When the samples contained carnosine that was already mixed with the media, the added BPEI-coated IONRs had no influence on the regular effect of carnosine, in all conditions. The BPEI-coated IONRs’ existence in the same vicinity of carnosine in vitro did not mask the treatment because the carnosine stayed free (Figure 17B). The presented results show that cationic BPEI-coated IONRs are promising in terms of their intracellular delivery into brain tumour cells, which is similar to that of colon cancer cells [63]. The fast uptake and localisation of BPEI-coated IONRs in the nucleus was used in drug delivery for GBM gene therapy or in the ca. 20-fold enhancement of the cell entry of hydrophobic drugs [1,69]. Adding the mild hyperthermia treatment gives the advantage of controlled released carnosine from the IONRs to the mitochondria. Thus, the presence of a safe amount of heated iron oxide in the same vicinity as the encapsulated drug could lead to long term GBM therapy, and to improve the safety profile of the astrocytes’ viability without alterations in blood–brain barrier permeability [71].

## 4. Conclusions

We have successfully demonstrated that carnosine inhibits both the growth and mobility of U87 glioblastoma brain tumour cells in vitro. Loading the carnosine into a sponge-like dendritic polymer matrix, in the form of branched polyethyleneimine, significantly reduces the risk of the drug being metabolised, and offers a stable environment from which the drug can be released at the site of action. The carnosine is readily released from the polymer matrix upon mild hyperthymia, without any degradation to the dipeptide or healthy cells. The addition of superparamagnetic iron oxide nanorods to the carrier not only enables it to be traced via MRI but can also be used to induce localised hyperthymia in the presence of an oscillating magnetic field [72]. This will be the subject of our continued studies in this area, alongside an animal model.

In conclusion, we have demonstrated a proof of concept that carnosine can be successfully loaded into a nanometre scale carrier that protects, localises, and releases the active ingredient via an external stimulus. It thereby provides a potential complimentary theragnostic, alongside traditional surgical techniques, towards the holistic treatment of glioblastoma brain tumours.

## Data Availability

The data presented in this study are available on request from the corresponding author.

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
