# Peer review of "Synthesis and Functionalisation of Superparamagnetic Nano-Rods towards the Treatment of Glioblastoma Brain Tumours"

_nanomaterials, 2021, doi:10.3390/nano11092157_

Round 1
Reviewer 1 Report
Review of the manuscript „Synthesis and Functionalisation of Superparamagnetic Nano-Rods Towards the Treatment of Glioblastoma Brain Tumours” by Habra et al..
Thank you for giving me the opportunity to review this indeed interesting manuscript. The authors present work on synthesizing magnetic iron oxide nanorods with a BPEI-coating and embedded carnosine, a natural dipeptide, which is thought to have anticancer effects. Release of this active ingredient should be realized by mild hyperthermia.
All in all the manuscript and the underlying work is mostly sound and well done, but some important information or experiments are missing and in some parts it seems the authors are presenting their knowledge about nanoparticles in a quite confusing way. Therefore, the manuscript shows a number of weaknesses that need to be improved before it can be considered for publication after major revisions.
In the following, I will first address general things to improve and then specifically chapter by chapter.
Language:
In general the manuscript is well written and mostly well understandable. Besides this, the authors are using the expression “vector” for the nanorods, which is a rather unusual designation to my taste. Do they want to avoid “carrier”, which would be more common? But there are a number of typos and obviously missing words. This should not occur in general but is even more astonishing having in mind that some of the authors are native speakers. It gives a bit the impression of a not thoroughly enough prove reading and subsequently a suspicious mind might apply this also to the data handling. Therefore, I would strongly recommend avoiding this.
Examples:
page 2, line 110: After “ultracentrifugation” the values in the brackets are not given but only “time RPM”.
page 2, line 69: There is a full stop after the citation.
page 4, line 193: The authors do not tell the reader to which element or substance the concentrations of 0, 1, 5 and µM refer to. Is it iron?
page 6, line 268: use vortexing
page 7, line 300: I guess the authors wanted to write “black powder” instead of “power”.
page 9, line 346: I would suggest: “Therefore, colloidal stability of SPIONs commonly are demonstrated…”
page 9, line 378: “significant effect” instead of “significant affect”.
Page 10, line 398: “HEPES” instead of “HEPS”?
Page 10, line 401: “distilled” instead of “Distilled”.
Page 11, line 418: “every hour” instead of “ever hour”.
Page 12, line 490 and 495: “IncuCyte®” instead of “IncuCyte (R)”.
Page 17, line 613: “until” instead of “util”.
Page 17, line 615: “U87” instead of “U78”.
Headings:
Sometimes the second noun is written with a first capital letter and sometimes not.
Images:
The images are well done but the labeling is a mess. In most figures the labeling is so small that one only has the chance to read them, by magnifying the document. Additionally, the sizing of the labeling is different between the figures, but sometimes even in one figure itself.
Therefore I recommend to reformatting all figures and standardizing the labeling.
Summary:
No concerns despite of the use of “…superparamagnetic iron oxide…”, which is not shown in the results and the missing hint that there is obviously no carnosine release and effect under normal cell culture conditions but only at 40°C.
Chapter 1 “Introduction”:
The Introduction is well balanced and leading to the point of the manuscript. Nevertheless, there are some statements that could be controversial or capable of being misunderstood.
From page one, line 44 to page 2, line 51 the authors claim that SPIONs are emerging as MRI contrast agents because of their lower toxicity compared to gadolinium. This is somehow a bit simplified. If you are comparing pure gadolinium or gadolinium ions not tightly bound in a chemical complex it is indeed very toxic. Compared to free gadolinium, iron oxide particles and iron complexes show really low toxicity. But if one injects iron salts like FeCl2 intravenously, one can induce blood clots. Iron oxide nanoparticles as well as iron complexes can induce severe reactions that can lead to death in rare cases (please see FDA release to Ferumoxytol). This can happen in human beings and pigs (Fülöp et al. 2018). This is the reason, why clinicians are very conservative in applying intravenously iron as supplement in anemia and such infusions have to be given really slowly. Mice and rats seem to be not so susceptible for this.
Next I do not really understand, what the authors want to tell the reader in the passage between page 2 line 58 and 62. First the authors claim that SPIONs in a size range (please add “size” in that passage, if it stays there) of 20-150nm are reported to have a good renal clearance, which as far as I do understand means that they are quickly removed from blood by the kidneys. But this is far too big for being filtered by the kidneys. I have never encountered SPIONs in the kidneys of animals. As far as I informed substances filtered by the kidneys should be smaller than 5nm. Usually SPIONs are to be found in the liver or, when they are taken up by macrophages in the lymph nodes and spleen. Sometimes, they can also get stuck in the lung. This all can be controlled by a suitable coating. This is true.
In the same sentence the authors tell the reader that other SPIONs avoid opsonisation, which does mean that immunoglobulines and factors of the immune system do not bind at their surface and hence uptake of these particles into immune cells would be attenuated. This would then lead to a higher blood half live, which could be – in my opinion – favorable for a contrast agent and/or a drug carrier. But then – again in the same sentence – the authors state that uptake into macrophages and thereby removal from the blood stream can be accelerated by controlling the surface charge via the addition of a surface coating.
In my opinion this whole passage is a total mess and reading this I am not sure, if the authors have a good understanding about blood stability and blood compatibility of SPIONs. After my experience, uncoated nanoparticles cannot be administered intravenously, because they immediately agglomerate and precipitate. Therefore, a suitable coating has to be added for providing colloidal stability. Indeed, this can be modified by the surface charge. But as depicted before, for being effective in vivo as a drug carrier or MRI-contrast agent, a prolonged blood half live is inevitable. If this is not the case the particles and the drug will quickly end up in the organs I mentioned before unless there is a burst release of the drug in the first minutes after application.
Nevertheless I agree with the authors that colloidal stability in cell culture media and bodily fluids is one of the biggest challenges in the development of metallic but also all other nanoparticles.
So, I would advise the authors to frame this passage a bit more carefully and structured and to really look carefully in the literature they present. As I said, I do not believe in SPIONs of that size removed by the kidneys.
Chapter 2 “Materials and Methods”:
This part is well written and understandable. It is impressing how many high performance analyses were done to characterize the nanorods. I found a few typos but recommend to check thoroughly again. But what I am missing here is how the carnosine loading was done and analyzed.
Page 5, lines 212 and 213: I have the impression that the first sentence of chapter 2.4.3.2 should be the last of chapter 2.3.4.1.
Chapter 3 “Results and Discussion”:
Chapter 3.1 “Synthesis and Characterization” is widely a repetition of the corresponding chapters in the “Materials and Methods” part. Here now comes a little description of the carnosine binding. In my opinion this chapter should be straightened and redundancies with chapter 2 removed as far as possible. The part with the carnosine binding could be pushed into the “Methods” chapter. Just the result of this binding should be in 3.1 and how much carnosine could be bound e.g. in µg/ml or any other reasonable concentration.
I do not find Figure S2 referenced in the text.
Major points here:
The authors claim to synthesize nanoparticles that exhibit superparamagnetic behavior. Since this is a very delicate characteristic, I would advise the authors using vibrating sample magnetometry (VSM) or a similar technique to show superparamagnetism of the IONRs.
Secondly, I am missing a clear depiction of the loading capacity of carnosine onto the nanorods e.g. in µg/ml.
And subsequently, I am missing completely the release kinetics of carnosine with and without mild hyperthermia. If the authors do not show release – and the cell culture experiments do not count for this – they cannot be sure, if the effect is from released carnosine or carnosine still caged in the sponge like surface coating on the nanoparticles. Another possibility could be that the BPEI and the carnosine detach together.
Chapter 3.2 “Colloidal Stability”
First the authors show the effect of different solutions on the colloidal stability. They are using DLS for testing stability in water up to 30 days. I am not quite sure, why they are investigating this at pH3 but it may be that they want to show stability against acidic conditions, too, or because some of the synthesis is done at that pH value. (Please explain in the manuscript.)
But I cannot imagine what this has to do with storage or in vivo conditions.
Nevertheless, it is nice to see that the nanorods do not change size and polydispersity over this time period.
I further do not understand why the authors come to the conclusion that colloidal stability in water for 30 days is essential for in vivo application. (Please explain in the manuscript.)
Usually, for this colloidal stability has to be shown in body fluids like blood, plasma or serum or at least in simulated body fluids.
Page 10, line 380: I do not understand what is meant with R2 on and why the value below 0.5 has an insignificant effect on the hydrodynamic diameter and polydispersity.
Page 10, line 380: A "significant change" in general means that there was a statistical analysis of the results. Please indicate the corresponding p-values. Please also repeat the original diameters and PD-values so that the reader can judge the values without the need of searching them in the text.
One could write: “…the hydrodynamic diameter changed from …. to …. and the polydispersity index changed from …. to ….”.
However, in figure 4 I cannot see a big change.
Page 10, line 387: In the passage before the authors state that the hydrodynamic size and the polydispersity index are changing considerably. I do not understand term “consequently”. Why does there occur no precipitation because the diameter and PI are changing so much?
Figure 5: The labeling, especially of the axes is too small. The text in the picture is hardly readable.
Page 10, line 396 to page 11, line 430: Much of this passage should be put into Chapter 2. The explanation is important but it does shift the attention of the reader away from the results of this interesting method of assessing colloidal stability in salt solutions.
Taken together 3.2 is much too long and describing a very complicated method for showing colloidal stability of the IONRs in water and a quite good one after 1 h in HEPES buffer and HEPES containing OptiMEM cell culture medium supplemented with 10% FCS. Nevertheless, quite an effect can be seen after 48h. Therefore, a measurement after 24h would have been very informative and I would recommend showing this.
Chapter 3.3 “Proliferation and Cytotoxicity”
The cell culture assays in this chapter were done very thoroughly and everything seems to be sound and plausible but again, some of the text could be in Chapter 2 or is redundant.
Page 12, line 484: I think the concentration of 125mM should be mentioned in this sentence.
Figure 7: - The Text in the pictures is not readable.
- I would recommend putting boxes around the pictures and the corresponding concentration of carnosine.
- Maybe the figure could be put at the very top of the page.
Figure 8: The labeling could be bigger.
Chapter 3.4 “Cell Migration and Invasion”
Using mitomycin C (mitoC) to discriminate between a carnosine effect on proliferation and migration seems to be a good attempt at the first sight. But I do not understand why the same concentrations of carnosine do have an effect on proliferation in the absence of mitoC but not when this drug is added to the cell culture medium. As far as I understand carnosine has an effect on the cell metabolism, thereby slowing down proliferation and – I guess – all other cell functions including migration. This effect rapidly starts at concentrations of 100mM and above. Why is this not the case when the DNA is cross linked?
Maybe I miss the point. I am sorry for that. To be honest this paragraph confuses me. Perhaps, the authors could try to explain this a bit more in detail.
Finally, what does this all tell the reader about invasion, which is somewhat more than just migration?
Chapter 3.5 “Cell Viability with IONRs”
Unlike all the other chapters this important part is quite short.
Chapter 3.6 “Uptake and Localization”
Figure 13: - What does NRS mean? I could not find it.
- the text and the labeling in (F) is hardly readable and the (F) seems to be overlaid by
the y-axis.
Page 17, line 597ff: Why do the authors mention ROS here and not in the chapter above?
Page 17, line 613:
I do not see data showing carnosine being transferred to daughter cells but only one cell.
I further do not see pictures done only with carnosine without the IONRs. At least the figure legends (Fig. 13 and 14) do not mention this.
I do not see carnosine crystals in mitochondria.
For all this, there should be pictures without any treatment, carnosine, the IONRs and carnosine loaded IONRs.
Figure 14: the text in the figure is hardly readable.
Figure 15: The color of the scale bars makes them hard to read as well as the size. The scale bar sizes are a mess. Please use same sizes.
Please use another color than yellow for the text. It is hard to read, too.
Chapter 3.7 “The effect of (NR/carnosine) synergism in controlled release”
As mentioned before, the authors do not tell us in the something about loading capacity and release in numbers. I am missing the concentration of the IONRs in the text. This vital information is only visible in Figure 16 at high magnification.
Fig. 16:
- The labeling is far too small and not readable without magnification.
- What does the labeling Abs of the y-axis mean?
- A is not centered in its box.
- The carnosine concentration is missing.
Nonetheless, I agree that there is an effect of the combination of IONRs, carnosine and mild hyperthermia of 40°C, which might be due to carnosine release. Nevertheless Figure 16 does not show a dramatic effect of the used carnosine concentration, which itself is not mentioned, alone or in combination with the IONRs.
Therefore, I am not sure, if the effect will be much better in a living organism and would advise the authors to write their conclusions throughout the whole text a bit more defensive.
Taken together, the authors conducted an extensive study in characterizing the newly developed BPEI-IONRs and the effect of carnosine and BPEI-IONRs loaded with carnosine but left out important information and experiments. Therefore, I am afraid there is still a lot work to do for bringing this study on a status that can be published in “Nanomaterials”.
References:
Tamás Fülöp, Réka Nemes, Tamás Mészáros, Rudolf Urbanics, Robbert Jan Kok, Joshua A. Jackman, Nam-Joon Cho, Gert Storm, János Szebeni: Complement activation in vitro and reactogenicity of low-molecular weight dextran-coated SPIONs in the pig CARPA model: Correlation with physicochemical features and clinical information; Journal of Controlled Release; Volume 270; 2018; Pages 268-274; ISSN 0168-3659,

Author Response
The authors would like to thank the Reviewer for their comments. Alongside the resubmitted manuscript, please see our responses to their questions and suggestions:
Language:
In general the manuscript is well written and mostly well understandable. Besides this, the authors are using the expression “vector” for the nanorods, which is a rather unusual designation to my taste. Do they want to avoid “carrier”, which would be more common? But there are a number of typos and obviously missing words. This should not occur in general but is even more astonishing having in mind that some of the authors are native speakers. It gives a bit the impression of a not thoroughly enough prove reading and subsequently a suspicious mind might apply this also to the data handling. Therefore, I would strongly recommend avoiding this.
We’ve now either removed the term “vector” or substituted it with “carrier” as suggested.
The following typos have also been amended in the main sript:
page 2, line 110: After “ultracentrifugation” the values in the brackets are not given but only “time RPM”.
page 2, line 69: There is a full stop after the citation.
page 4, line 193: The authors do not tell the reader to which element or substance the concentrations of 0, 1, 5 and µM refer to. Is it iron?
page 6, line 268: use vortexing
page 7, line 300: I guess the authors wanted to write “black powder” instead of “power”.
page 9, line 346: I would suggest: “Therefore, colloidal stability of SPIONs commonly are demonstrated…”
page 9, line 378: “significant effect” instead of “significant affect”.
Page 10, line 398: “HEPES” instead of “HEPS”?
Page 10, line 401: “distilled” instead of “Distilled”.
Page 11, line 418: “every hour” instead of “ever hour”.
Page 12, line 490 and 495: “IncuCyte®” instead of “IncuCyte (R)”.
Page 17, line 613: “until” instead of “util”.
Page 17, line 615: “U87” instead of “U78”.
Headings:
Sometimes the second noun is written with a first capital letter and sometimes not.
All headings are now consistent throughout.
Images:
The images are well done but the labeling is a mess. In most figures the labeling is so small that one only has the chance to read them, by magnifying the document. Additionally, the sizing of the labeling is different between the figures, but sometimes even in one figure itself.
Therefore I recommend to reformatting all figures and standardizing the labeling.
The quality of the images and corresponding text have been improved to improve readably.
Summary:
No concerns despite of the use of “…superparamagnetic iron oxide…”, which is not shown in the results and the missing hint that there is obviously no carnosine release and effect under normal cell culture conditions but only at 40°C.
The use of the term superparamagnetic is discussed later.
Chapter 1 “Introduction”:
The Introduction is well balanced and leading to the point of the manuscript. Nevertheless, there are some statements that could be controversial or capable of being misunderstood.
From page one, line 44 to page 2, line 51 the authors claim that SPIONs are emerging as MRI contrast agents because of their lower toxicity compared to gadolinium. This is somehow a bit simplified. If you are comparing pure gadolinium or gadolinium ions not tightly bound in a chemical complex it is indeed very toxic. Compared to free gadolinium, iron oxide particles and iron complexes show really low toxicity. But if one injects iron salts like FeCl2 intravenously, one can induce blood clots. Iron oxide nanoparticles as well as iron complexes can induce severe reactions that can lead to death in rare cases (please see FDA release to Ferumoxytol). This can happen in human beings and pigs (Fülöp et al. 2018). This is the reason, why clinicians are very conservative in applying intravenously iron as supplement in anemia and such infusions have to be given really slowly. Mice and rats seem to be not so susceptible for this.
We’ve address this comment in the main text and added and additional reference. Lines 51-53
Next I do not really understand, what the authors want to tell the reader in the passage between page 2 line 58 and 62. First the authors claim that SPIONs in a size range (please add “size” in that passage, if it stays there) of 20-150nm are reported to have a good renal clearance, which as far as I do understand means that they are quickly removed from blood by the kidneys. But this is far too big for being filtered by the kidneys. I have never encountered SPIONs in the kidneys of animals. As far as I informed substances filtered by the kidneys should be smaller than 5nm. Usually SPIONs are to be found in the liver or, when they are taken up by macrophages in the lymph nodes and spleen. Sometimes, they can also get stuck in the lung. This all can be controlled by a suitable coating. This is true.
We’ve added a reference to a review article that helps support the original text for clarity. Line 65
In the same sentence the authors tell the reader that other SPIONs avoid opsonisation, which does mean that immunoglobulines and factors of the immune system do not bind at their surface and hence uptake of these particles into immune cells would be attenuated. This would then lead to a higher blood half live, which could be – in my opinion – favorable for a contrast agent and/or a drug carrier. But then – again in the same sentence – the authors state that uptake into macrophages and thereby removal from the blood stream can be accelerated by controlling the surface charge via the addition of a surface coating.
In my opinion this whole passage is a total mess and reading this I am not sure, if the authors have a good understanding about blood stability and blood compatibility of SPIONs. After my experience, uncoated nanoparticles cannot be administered intravenously, because they immediately agglomerate and precipitate. Therefore, a suitable coating has to be added for providing colloidal stability. Indeed, this can be modified by the surface charge. But as depicted before, for being effective in vivo as a drug carrier or MRI-contrast agent, a prolonged blood half live is inevitable. If this is not the case the particles and the drug will quickly end up in the organs I mentioned before unless there is a burst release of the drug in the first minutes after application.
Nevertheless I agree with the authors that colloidal stability in cell culture media and bodily fluids is one of the biggest challenges in the development of metallic but also all other nanoparticles.
So, I would advise the authors to frame this passage a bit more carefully and structured and to really look carefully in the literature they present. As I said, I do not believe in SPIONs of that size removed by the kidneys.
Hopefully the additional reference (17) help clarify this to the reader and doesn’t detract from the main studies that we have presented.
Chapter 2 “Materials and Methods”:
This part is well written and understandable. It is impressing how many high performance analyses were done to characterize the nanorods. I found a few typos but recommend to check thoroughly again. But what I am missing here is how the carnosine loading was done and analyzed.
Page 5, lines 212 and 213: I have the impression that the first sentence of chapter 2.4.3.2 should be the last of chapter 2.3.4.1.
This has been amended as suggested.
Chapter 3 “Results and Discussion”:
Chapter 3.1 “Synthesis and Characterization” is widely a repetition of the corresponding chapters in the “Materials and Methods” part. Here now comes a little description of the carnosine binding. In my opinion this chapter should be straightened and redundancies with chapter 2 removed as far as possible. The part with the carnosine binding could be pushed into the “Methods” chapter. Just the result of this binding should be in 3.1 and how much carnosine could be bound e.g. in µg/ml or any other reasonable concentration.
We have reworked this section and adopted the change or units.
I do not find Figure S2 referenced in the text.
This refers to an image in the supplementary data “S2A”
Major points here:
The authors claim to synthesize nanoparticles that exhibit superparamagnetic behavior. Since this is a very delicate characteristic, I would advise the authors using vibrating sample magnetometry (VSM) or a similar technique to show superparamagnetism of the IONRs.
The superparamagnetic properties of Fe3O4 are well documented and evident from the MRI studies described in the main text, as such we believe that a detailed magnetometry investigation is beyond the scope of this body of work.
Secondly, I am missing a clear depiction of the loading capacity of carnosine onto the nanorods e.g. in µg/ml.
And subsequently, I am missing completely the release kinetics of carnosine with and without mild hyperthermia. If the authors do not show release – and the cell culture experiments do not count for this – they cannot be sure, if the effect is from released carnosine or carnosine still caged in the sponge like surface coating on the nanoparticles. Another possibility could be that the BPEI and the carnosine detach together.
This is now discussed in further detail in section 3.6, including Figs 16 and S12.
Chapter 3.2 “Colloidal Stability”
First the authors show the effect of different solutions on the colloidal stability. They are using DLS for testing stability in water up to 30 days. I am not quite sure, why they are investigating this at pH3 but it may be that they want to show stability against acidic conditions, too, or because some of the synthesis is done at that pH value. (Please explain in the manuscript.)
But I cannot imagine what this has to do with storage or in vivo conditions.
Nevertheless, it is nice to see that the nanorods do not change size and polydispersity over this time period.
We’ve added a sentence at the end of this paragraph to elaborate on this. Lines 454-458
I further do not understand why the authors come to the conclusion that colloidal stability in water for 30 days is essential for in vivo application. (Please explain in the manuscript.)
Usually, for this colloidal stability has to be shown in body fluids like blood, plasma or serum or at least in simulated body fluids.
This study was conducted in order to evaluate the “shelf-life” of the active not it’s physiological stability.
Page 10, line 380: I do not understand what is meant with R2 on and why the value below 0.5 has an insignificant effect on the hydrodynamic diameter and polydispersity.
The R2 correlation coefficient is a statistic indicator that helps to predict the outcome of our hypothesis that, there is an increase in size with time. Therefore, an R2 closer to zero helps us to reject this.
Page 10, line 380: A "significant change" in general means that there was a statistical analysis of the results. Please indicate the corresponding p-values. Please also repeat the original diameters and PD-values so that the reader can judge the values without the need of searching them in the text.
One could write: “…the hydrodynamic diameter changed from …. to …. and the polydispersity index changed from …. to ….”.
However, in figure 4 I cannot see a big change.
The suggested change to the text has been adopted. Line447-448
Page 10, line 387: In the passage before the authors state that the hydrodynamic size and the polydispersity index are changing considerably. I do not understand term “consequently”. Why does there occur no precipitation because the diameter and PI are changing so much?
The word “consequently” has been removed and the text changed. Lines 449-458
Figure 5: The labeling, especially of the axes is too small. The text in the picture is hardly readable.
The legend has been amened to improve readability.
Page 10, line 396 to page 11, line 430: Much of this passage should be put into Chapter 2. The explanation is important but it does shift the attention of the reader away from the results of this interesting method of assessing colloidal stability in salt solutions.
The methodology section has be moved as suggested.
Taken together 3.2 is much too long and describing a very complicated method for showing colloidal stability of the IONRs in water and a quite good one after 1 h in HEPES buffer and HEPES containing OptiMEM cell culture medium supplemented with 10% FCS. Nevertheless, quite an effect can be seen after 48h. Therefore, a measurement after 24h would have been very informative and I would recommend showing this.
The requested additional data has now been included and discussed. Fig 5
Chapter 3.3 “Proliferation and Cytotoxicity”
The cell culture assays in this chapter were done very thoroughly and everything seems to be sound and plausible but again, some of the text could be in Chapter 2 or is redundant.
We have removed the superfluous text.
Page 12, line 484: I think the concentration of 125mM should be mentioned in this sentence.
corrected
Figure 7: - The Text in the pictures is not readable.
- I would recommend putting boxes around the pictures and the corresponding concentration of carnosine.
- Maybe the figure could be put at the very top of the page.
This has now been amended.
Figure 8: The labeling could be bigger.
This has now been amended.
Chapter 3.4 “Cell Migration and Invasion”
Using mitomycin C (mitoC) to discriminate between a carnosine effect on proliferation and migration seems to be a good attempt at the first sight. But I do not understand why the same concentrations of carnosine do have an effect on proliferation in the absence of mitoC but not when this drug is added to the cell culture medium. As far as I understand carnosine has an effect on the cell metabolism, thereby slowing down proliferation and – I guess – all other cell functions including migration. This effect rapidly starts at concentrations of 100mM and above. Why is this not the case when the DNA is cross linked?
Maybe I miss the point. I am sorry for that. To be honest this paragraph confuses me. Perhaps, the authors could try to explain this a bit more in detail.
Finally, what does this all tell the reader about invasion, which is somewhat more than just migration?
We’ve added an additional paragraph to help explain this. Lines 641-644
Chapter 3.5 “Cell Viability with IONRs”
Unlike all the other chapters this important part is quite short.
We’ve added some additional text to the main article and to the SI. Lines 671-677
Chapter 3.6 “Uptake and Localization”
Figure 13: - What does NRS mean? I could not find it.
NR has been change to IONR throughout.
- the text and the labeling in (F) is hardly readable and the (F) seems to be overlaid by
the y-axis.
Image has been expended to improve readability.
Page 17, line 597ff: Why do the authors mention ROS here and not in the chapter above?
This has been moved as suggested.
Page 17, line 613:
I do not see data showing carnosine being transferred to daughter cells but only one cell.
I further do not see pictures done only with carnosine without the IONRs. At least the figure legends (Fig. 13 and 14) do not mention this.
I do not see carnosine crystals in mitochondria.
For all this, there should be pictures without any treatment, carnosine, the IONRs and carnosine loaded IONRs.
Additional images of the control cells have been deposited in the SI.
Figure 14: the text in the figure is hardly readable.
Image has been expended to improve readability.
Figure 15: The color of the scale bars makes them hard to read as well as the size. The scale bar sizes are a mess. Please use same sizes.
Please use another color than yellow for the text. It is hard to read, too.
Image has been edited to improve readability.
Chapter 3.7 “The effect of (NR/carnosine) synergism in controlled release”
As mentioned before, the authors do not tell us in the something about loading capacity and release in numbers. I am missing the concentration of the IONRs in the text. This vital information is only visible in Figure 16 at high magnification.
This additional information has been added to section 3.6
Fig. 16:
- The labeling is far too small and not readable without magnification.
- What does the labeling Abs of the y-axis mean?
- A is not centered in its box.
- The carnosine concentration is missing.
Figures have been corrected and edited to improve readability.
Nonetheless, I agree that there is an effect of the combination of IONRs, carnosine and mild hyperthermia of 40°C, which might be due to carnosine release. Nevertheless Figure 16 does not show a dramatic effect of the used carnosine concentration, which itself is not mentioned, alone or in combination with the IONRs.
Therefore, I am not sure, if the effect will be much better in a living organism and would advise the authors to write their conclusions throughout the whole text a bit more defensive.
Taken together, the authors conducted an extensive study in characterizing the newly developed BPEI-IONRs and the effect of carnosine and BPEI-IONRs loaded with carnosine but left out important information and experiments. Therefore, I am afraid there is still a lot work to do for bringing this study on a status that can be published in “Nanomaterials”.
References:
Tamás Fülöp, Réka Nemes, Tamás Mészáros, Rudolf Urbanics, Robbert Jan Kok, Joshua A. Jackman, Nam-Joon Cho, Gert Storm, János Szebeni: Complement activation in vitro and reactogenicity of low-molecular weight dextran-coated SPIONs in the pig CARPA model: Correlation with physicochemical features and clinical information; Journal of Controlled Release; Volume 270; 2018; Pages 268-274; ISSN 0168-3659,
Reviewer 2 Report
The paper submitted for publication in “Nanomaterials” by K. Habra et al. is related to the synthesis, characterization of a superparamagnetic iron oxide nano-rods (IONRs) and, evaluation of their biological effects on glioblastoma tumors. The work is sound and presents a set of very well conducted and interesting biological studies, however in my opinion that some issues must be addressed.
- Are these IONRs and the active species selective? What is the effect on non-tumor cells?
- For me, the releasing effect that the authors claim is not clear. There is no guarantee that the observed results are due to release, there may be some synergistic effect which is activated by the increase in temperature. The authors should carry out analytical studies that effectively demonstrate this release over time and for both temperatures (37° and 40°).
- Typos:
Line 92 – define PEI
Line 110 - ultra-centrifugation: please indicate the time and rpm
BPEI - define the first time it appears in the text (it is defined only in section 3.1)
Author Response
Are these IONRs and the active species selective? What is the effect on non-tumor cells?
We’ve expanded section 3.5 (and the SI) to highlight that they are non-selective at this point in our studies. We’ve also included several new references to support this discussion.
For me, the releasing effect that the authors claim is not clear. There is no guarantee that the observed results are due to release, there may be some synergistic effect which is activated by the increase in temperature. The authors should carry out analytical studies that effectively demonstrate this release over time and for both temperatures (37° and 40°).
Section 3.6 has been expanded to induced this additional information.
The following typos have been corrected:
Line 92 – define PEI
Line 110 - ultra-centrifugation: please indicate the time and rpm
BPEI - define the first time it appears in the text (it is defined only in section 3.1)
Round 2
Reviewer 2 Report
The review was in line with my comments, therefore, I recommend the manuscript for publication in its current form.